# RN-Net: A Deep Learning Approach to 0–2 Hour Rainfall Nowcasting Based on Radar and Automatic Weather Station Data

**DOI:** 10.3390/s21061981

**Published:** 2021-03-11

**Authors:** Fuhan Zhang, Xiaodong Wang, Jiping Guan, Meihan Wu, Lina Guo

**Affiliations:** 1School of Computer, National University of Defense Technology, Changsha 410000, China; zhangfuhan19@nudt.edu.cn (F.Z.); meihanwu20@nudt.edu.cn (M.W.); guolina19@nudt.edu.cn (L.G.); 2School of Meteorology and Oceanography, National University of Defense Technology, Changsha 410000, China; guan_ji_ping@126.com

**Keywords:** deep learning, RNN, rainfall nowcasting, radar echo data, automatic weather stations data

## Abstract

Precipitation has an important impact on people’s daily life and disaster prevention and mitigation. However, it is difficult to provide more accurate results for rainfall nowcasting due to spin-up problems in numerical weather prediction models. Furthermore, existing rainfall nowcasting methods based on machine learning and deep learning cannot provide large-area rainfall nowcasting with high spatiotemporal resolution. This paper proposes a dual-input dual-encoder recurrent neural network, namely Rainfall Nowcasting Network (RN-Net), to solve this problem. It takes the past grid rainfall data interpolated by automatic weather stations and doppler radar mosaic data as input data, and then forecasts the grid rainfall data for the next 2 h. We conduct experiments on the Southeastern China dataset. With a threshold of 0.25 mm, the RN-Net’s rainfall nowcasting threat scores have reached 0.523, 0.503, and 0.435 within 0.5 h, 1 h, and 2 h. Compared with the Weather Research and Forecasting model rainfall nowcasting, the threat scores have been increased by nearly four times, three times, and three times, respectively.

## 1. Introduction

Precipitation is the main forecast element of Numerical Weather Prediction [1] (NWP), which has an important impact on people’s daily life [2,3] and disaster prevention and mitigation [4]. After years of development, the current short-term and medium-term NWP models have become more and more accurate. However, for rainfall nowcasting, it is difficult to give accurate forecast results due to spin-up [5] and other problems in NWP models.

In recent years, artificial intelligence has become the new engine of the global scientific and technological revolution, and some scholars have applied machine learning and deep learning to precipitation forecasting [6,7,8,9,10,11,12,13]. After Shi et al. [14] achieved precipitation intensity nowcasting by radar echo extrapolation, it has emerged as a hot research topic in the meteorological community. They formulated radar echo extrapolation as a spatiotemporal prediction problem, and used ConvLSTM applying convolution structure to LSTM to predict future radar echo data by past radar echo data. They then used the Z–R relationship to convert predicted radar echo data into precipitation intensity data to realize precipitation intensity nowcasting. They conducted experiments on the dataset composed of radar echo data during the 97 days of precipitation in Hong Kong in 2011–2013. ConvLSTM has reached 0.577 for the Critical Success Index (CSI) with a threshold of 0.5 mm/h in the next 1.5 h, which has a strong ability to forecast precipitation intensity. In 2017, Shi et al. [15] further proposed TrajGRU to improve the effect of precipitation intensity nowcasting. It used the generated optical flow [16] to realize a connection structure based on position changes, and the point in the convolution structure is connected to points with higher correlation instead of a fixed number of surrounding points. In the experiment on the HKO-7 dataset, TrajGRU’s CSI reached 0.552 in the next 2 h with a threshold of 0.5 mm/h.

After Shi et al. [14] formulated radar echo extrapolation as a spatiotemporal prediction problem, many spatiotemporal prediction methods [17,18] regarded radar echo extrapolation as one of the problems to evaluate the effectiveness of their methods. Wang et al. [17] proposed a spatiotemporal prediction method celled PredRNN, which solves the problem of spatial features of each layer of ConvLSTM being independent of each other in time series. Spatiotemporal memory units are added to PredRNN and connected through a zigzag structure so that features can be propagated both spatially and temporally. They conducted experiments on the radar echo dataset in Guangzhou, and the mean square error of PredRNN is 30% higher than that of ConvLSTM. Bonnet et al. [19] applied the spatiotemporal prediction method PredRNN++ [20] to the precipitation intensity nowcasting. PredRNN++ utilized the Causal LSTM unit to integrate temporal and spatial features and Gradient Highway (GHU) that could alleviate gradient disappearance. However, the precipitation intensity nowcasting based on radar echo extrapolation has two main problems. One is that the radar echo data cannot reflect the real-world distribution of precipitation, which is caused by the working principle of the radar and various noises. The second is that the precipitation intensity converted from the radar echo data are inconsistent with the actual precipitation intensity, which is caused by the inaccurate Z–R relationship.

Compared with radar echo data used in precipitation intensity nowcasting, the rainfall data used in rainfall nowcasting can be directly measured by rain gauges and other equipment, which can more accurately reflect the real-world precipitation. Currently, there are few rainfall nowcasting methods based on machine learning and deep learning. Zhang et al. [21] used a multi-layer perceptron to forecast the rainfall data of 56 weather stations in China of the next 3 h. The forecast was derived from 13 physical factors related to precipitation in the surrounding area. Although the forecast results can meet the needs of nowcasting, its spatial resolution is too low to achieve large-area rainfall nowcasting.

Existing rainfall nowcasting methods based on machine learning and deep learning are difficult in order to achieve rainfall nowcasting with high spatiotemporal resolution. To be able to achieve this, we set the grid data interpolated from the rainfall data in the dense automatic weather station as the forecast object. Since the original data are directly measured by the rain gauge, the grid data will reflect the real-world rainfall distribution as much as possible. In the grid data, the forecasting time resolution and spatial resolution are 30 min and 5 km, respectively, which can meet the high spatiotemporal resolution requirement. As the forecast target is sequential grid data, we formulate this forecasting problem as a spatiotemporal prediction problem, which predicts future development through past spatiotemporal features [22]. Rainfall depends on rainfall intensity and rainfall duration so that its evolution is more complicated and diverse. Therefore, we utilize both rainfall data and radar echo data as our input data to gain more meteorological spatiotemporal features which can support this complex forecasting.

Based on experiments with multiple models, we propose a dual-input dual-encoder RNN, namely Rainfall Nowcasting Network (RN-Net). RN-Net extracts spatiotemporal features of the rainfall and the radar echo data via dual encoders. Then, these features are combined by a fusion module. Finally, the fused features are fed into a predictor to make forecasts. In order to reasonably evaluate the effectiveness of rainfall nowcasting, we propose a new performance metric that combines multiple metrics in the field of meteorological and spatiotemporal prediction. In the experiment, 10 months of radar echo data and rainfall data in the southeastern coastal area of China were used as deep learning samples and compared with rainfall nowcasting of the Weather Research and Forecasting (WRF) [23] model. The results are expected to provide convenience for daily activities such as travel and irrigation, and provide a basis for early warning of natural disasters such as floods and mudslides.

The rest of this paper is organized as follows: Section 2 introduces the preparatory work. Section 3 details the proposed RN-Net framework. Experimental results are demonstrated in Section 4. Finally, we conclude this paper and put forward some suggestions for future work.

## 2. Preliminary

### 2.1. Data Details

The radar echo data used in this article is Doppler radar mosaic data. Radar echo data contains various echo noise, such as non-meteorological echo, interference echo, etc., which mislead prediction. Therefore, we construct a singular point filter and a bilateral filter to filter the value domain and the spatial domain, which can effectively eliminate the pulsation and clutter while retaining the echo characteristics. In addition, a high-pass filter is constructed to remove data below 15 dBZ, and only data related to precipitation are retained. Since the data will be saved as an image format, we convert the radar echo data into pixel data.

The rainfall interpolation data of the automatic weather station is selected as the rainfall data in this article. Automatic weather stations are widely distributed and the distribution of automatic weather stations used in this paper is shown in Figure 1a. Rainfall data are usually measured by rain gauges. It collects the rainfall in a specific area and divides the rainfall volume by the surface area to obtain the depth of rainfall. Inspired by the E-OBS dataset [24], we interpolate the rainfall point data into a uniform grid. We use Inverse Distance Weight [25] (IDW) to interpolate the rainfall data of 13,655 automatic weather stations in the forecasting area into a 240 × 240 grid. With such high-density data interpolation, the actual rainfall distribution is restored as much as possible. Figure 1b,c shows the effect of interpolation. IDW takes the distance between the interpolation point and the sample point as the weight for the weighted average. The closer the sample point is to the interpolation point, the greater the weight. The critical equation is as follows:(1)$$d_{ij}=\sqrt{{\left(x_{j}-x_{i}\right)}^{2}+{\left(y_{j}-y_{i}\right)}^{2}}$$
(2)$$\lambda_{ij}=\frac{\frac{1}{d_{ij}}}{\sum_{i=1}^{n}\frac{1}{d_{ij}}}$$
(3)$$Z\left(x_{j},y_{j}\right)=\sum_{i=1}^{n}\lambda_{ij}Z\left(x_{i},y_{i}\right)$$
where *n* is the number of selected sampling points closest to the interpolation point, which is set to 16 in the experiment. $\left(x_{i},y_{i}\right)$ represents the coordinates of the sample point, and $\left(x_{j},y_{j}\right)$ denotes the coordinates of the interpolation point. Z(·,·) is the value of this coordinate, and $d_{ij}$ is the distance between the sample point and the interpolation point. $\lambda_{ij}$ is the weight of the sample point to the interpolation point. Finally, we convert the interpolated rainfall data into pixel data.

In addition, the WRF model is used to compare the 0–2 h rainfall nowcasting effect of RN-Net. The WRF model [26] is configured with a one-domain nested grid system. The horizontal resolution of the domain is 5 km, with the grid points 240 × 240. The domain has 35 vertical layers, with the model top at 50 hPa. The boundary conditions are updated every 6 h from the 0.25° × 0.25° National Centers for Environmental Prediction (NCEP) Final Operational Model Global Tropospheric Analysis. The main physical parameterization schemes are shown in Table 1. The model is integrated every 6 h, the forecast time is 12 h, and the results are output every 30 min.

### 2.2. Problem Definition

Our goal is to forecast future automatic weather station rainfall interpolation data by past radar echo data and rainfall data. We formally define this problem as follows: suppose the current moment is t=0. We have access to the radar echo data ${\left[RE_{t}\right]}_{t=-n}^{0}$ and the recent rainfall data ${\left[RF_{t}\right]}_{t=-m}^{0}$. Our task is to predict ${\left[{\hat{RF}}_{t}\right]}_{t=1}^{s}$, and make them as close as possible to ${\left[RF_{t}\right]}_{t=1}^{s}$, which is the real rainfall data for next time. Specifically, our goal is to find a mapping *f* such that
(4)$$\begin{array}{c} \overset{min}{\underset{f}{}}\;loss\left({\left[{\hat{RF}}_{t}\right]}_{t=1}^{s},{\left[RF_{t}\right]}_{t=1}^{s}\right)\;\;\\ s.t.\;{\left[{\hat{RF}}_{t}\right]}_{t=1}^{s}=f\left({\left[RE_{t}\right]}_{t=-n}^{0},{\left[RF_{t}\right]}_{t=-m}^{0}\right)) \end{array}$$

## 3. Method

### 3.1. Network Structure

In order to achieve high spatiotemporal resolution rainfall nowcasting, our model needs to obtain sufficient meteorological spatiotemporal features to support the forecast. Meanwhile, its RNN unit also needs to have stronger feature extraction and transmission capabilities. The network structure of RN-Net is shown in Figure 2.

Inspired by LightNet [27], RN-Net contains two encoders, a fusion module and a predictor. The time resolution of rainfall data from automatic weather stations is 30 min and the time resolution of radar echo data are 6 min. The time resolution difference between the two data are too large. Therefore, the two types of data cannot be encoded by the same encoder. RN-Net uses radar echo encoder and rainfall encoder to respectively encode the two kinds of data to generate spatiotemporal features. The fusion module composed of CNN is used to fuse the spatiotemporal features of the two data. Finally, the spatiotemporal features are input to the predictor, and the forecasting of future rainfall is output. We detail each component as follows.

**Radar Echo Encoder or Rainfall Encoder:** Both of these two encoders have the same network structure and parameters. The encoder has a three-layer structure, and each layer is composed of a layer of RNN and downsample unit. The downsample unit helps the model understand the high-level spatial features of the input data, so as to better extract the spatiotemporal features. The input of the first layer is radar echo data or rainfall data (${\left[RE_{t}\right]}_{t=-n}^{0}$ or ${\left[RF_{t}\right]}_{t=-m}^{0}$). Then, the hidden features of each time point of this layer are input into the downsample unit of the next layer, and the hidden state of the last time point ($h_{1}^{RE}$ or $h_{1}^{RF}$) is used as the output of the encoder in this layer. The second and third layers continue the same process. The final output of the encoder is the hidden state of each layer, and the formula is expressed as follows: (5)$$\left[\begin{array}{c} h_{3}^{RE}\\ h_{2}^{RE}\\ h_{1}^{RE} \end{array}\right]=Radar-Echo-Encoder\left({\left[RE_{t}\right]}_{t=-n}^{0}\right)$$
(6)$$\left[\begin{array}{c} h_{3}^{RF}\\ h_{2}^{RF}\\ h_{1}^{RF} \end{array}\right]=Rainfall-Encoder\left({\left[RF_{t}\right]}_{t=-m}^{0}\right)$$

**Fusion Module:** The radar echo data contain rich meteorological features. Due to its various noises, it cannot accurately reflect the real-world rainfall distribution. The rainfall data of the automatic weather station reflect the actual rainfall distribution. To obtain accurate rainfall nowcasting, the hidden features of the two data are combined. The fusion module superimposes the hidden features of the two, and then deep fusion through CNN. Its formula is expressed as follows:(7)$$\left[\begin{array}{c} h_{3}^{fusion}\\ h_{2}^{fusion}\\ h_{1}^{fusion} \end{array}\right]=Fusion\left(\left[\begin{array}{c} h_{1}^{RF}\\ h_{1}^{RE} \end{array}\right],\left[\begin{array}{c} h_{2}^{RF}\\ h_{2}^{RE} \end{array}\right],\left[\begin{array}{c} h_{3}^{RF}\\ h_{3}^{RE} \end{array}\right]\right)$$

**Rainfall Predictor:** The structure of rainfall predictor is similar to an encoder. It is also a three-layer structure, and each layer is composed of a layer of RNN and upsample units. The difference is that two layers of CNN are added to the output part, which is more conducive to generating forecasting data $[{\hat{RF}}_{t}{]}_{t=1}^{s}$ by spatiotemporal features. When forecasting, the input data of the predictor are the fused spatiotemporal hidden features. The third layer expands the corresponding fused hidden state in the future period. Then, the hidden state at each time is input to the upsample unit to generate the lower-level spatial hidden state, which is input into the next layer of the predictor. The second and first layers continue the same process. Finally, two layers of CNN output rainfall nowcasting based on low-level spatiotemporal features. The formula is as follows:(8)$${\left[{\hat{RF}}_{t}\right]}_{t=1}^{s}=Predictor\left(\left[\begin{array}{c} h_{3}^{fusion}\\ h_{2}^{fusion}\\ h_{1}^{fusion} \end{array}\right]\right)$$

TrajGRU is the RNN unit used in RN-Net. It is improved based on ConvGRU and overcomes the problem that the connection structure between the memory states in other convRNNs is fixed. For input data, TrajGRU and ConvGRU both use convolution as the connection structure, which makes it possible to obtain the spatial features of the input data. For memory state, TrajGRU uses a structure generation network to dynamically generate the optical flow between states as the connection structure. Such a flexible connection structure can more efficiently learn complex motion patterns such as rotation and zooming in spatiotemporal features. The settings (the kernel size, channels and stride) of each component of RN-Net are detailed in Table 2.

In addition to RN-Net, we also try two other dual-input dual-encoder methods. When ConvLSTM is used as the RNN unit of RN-Net, the radar echo encoder/rainfall encoder and predictor need to transmit cells and hidden states simultaneously, and the fusion module needs to fuse the two features separately. When using PredRNN as the backbone network of RN-Net, the radar echo encoder/rainfall encoder and the predictor need to transmit cell, hidden state, and spatiotemporal memory at the same time, and spatiotemporal memory needs to be interspersed with zigzags in the network. The fusion module needs to fuse the three features separately.

### 3.2. Implementation Details

The proposed neural networks are implemented with Pytorch [28] and are trained end-to-end. All network parameters are initialized with a normal distribution. All models are optimized with *L*2 loss, and they are trained using the Adam optimizer [29] with a starting learning rate of 10${}^{-4}$. The training process is stopped after 40,000 iterations, and the batch size of each iteration is set to 4. The rainfall data and radar echo data normalized to the range of [0, 1] are used as network input data.

## 4. Experiment

In this part, we evaluate the proposed models on the Southeastern China dataset. In Section 4.1, we introduce the details of the dataset. In Section 4.2, we introduce a new rainfall nowcasting performance metric, which combines multiple evaluation metrics in the meteorological field and the spatiotemporal prediction field. Section 4.3 compares RN-Net with other methods, including eight deep learning methods and the WRF model. We visualize two representative examples for further analysis in Section 4.4. Our experimental platform uses Ubuntu16.04, 32 GB memory, and two Nvidia RTX 2080 GPUs.

### 4.1. Dataset

**Radar Echo Data:** We use the data from the southeast coast of China. The data are stored in a 240 × 240 grid, and its spatial resolution is 5 km. The time resolution is 6 min, and the time range includes May to September in 2018 and 2019.

**Rainfall Data:** The spatial range, spatial resolution, and time resolution of rainfall data from automatic weather stations are the same as those of radar echo data. Its original time resolution is 10 min; due to the small value and sparse spatial distribution after interpolation, its time resolution is converted to 30 min.

In the dataset, 207 days for training, 29 days for validation, and 57 days for testing. The data on some days were incomplete due to equipment failure or other reasons. Our task can be defined as nowcasting the rainfall data of the next 2 h, based on the rainfall data of the past 2 h and the radar echo data of the past 1 h.

### 4.2. Performance Metric

In our methods, the rainfall nowcasting with the high spatiotemporal resolution is formulated as a spatiotemporal prediction problem to solve. The forecast result is the cumulative rainfall interpolation data of four frames within 0.5 h in the next 2 h, which is compared with the actual automatic weather station rainfall interpolation data to evaluate the forecast effect. To make a reasonable evaluation, we define a new performance metric by combining various evaluation metrics in the field of meteorology and spatiotemporal prediction.

Commonly used metrics for rainfall nowcasting in the meteorological field include TS, probability of detection (POD) and false alarm rate (FAR). In the experiment, we choose to use the thresholds 0.25 mm, 1 mm, and 2.5 mm to calculate these metrics. The threshold setting refers to the rainfall level, and the corresponding relationship is shown in Table 3. In order to show the effect of rainfall nowcasting in the next 2 h, we take three time periods within 0.5 h, 1 h, and 2 h for evaluation. The forecast rainfall and actual rainfall in these three periods are accumulated and used as evaluation data. In the field of spatiotemporal prediction, the forecasting results are evaluated frame by frame. The evaluation result in a period of time is the average of the evaluation results of each frame in the period. Applying this idea to our method evaluation, and the multi-frame evaluation results within 1 h and 2 h are averaged, including the Critical Success Index (CSI), POD, and FAR.

In addition, since the data are all two-dimensional grid data and are saved in the image format, we introduce two metrics, MAE and MSE, which respectively calculate the L1 distance and L2 distance between the truth data and the forecast data.

CSI and TS have the same calculation formula. TS evaluates accumulated rainfall in the period, and CSI is the average value of multiple frame evaluations in the period. The following are the calculation equations for these six evaluation metrics:(9)$$MSE=\sum_{x=1}^{w}\sum_{y=1}^{h}{\left({\hat{RF}}_{xy}-RF_{xy}\right)}^{2}$$
(10)$$MAE=\sum_{x=1}^{w}\sum_{y=1}^{h}\left|{\hat{RF}}_{xy}-RF_{xy}\right|$$
(11)CSI/TS=NA/(NA+NB+NC)
(12)POD=NA/(NA+NC)
(13)FAR=NB/(NA+NB)

Here, *w* and *h* are the width and height of the rainfall data, respectively. ${\hat{RF}}_{xy}$ and $RF_{xy}$ are the forecast rainfall and truth rainfall in the coordinates (x,y). NA, NB, NC, and ND represent the number of true-positive, false-positive, false-negative, and true-negative grid points.

Finally, the performance metric includes five evaluation results of cumulative rainfall within 0.5 h, 1 h and 2 h, and average values of the first two frames within 1 h and four frames within 2 h. Such performance metric not only reflect the forecasting effect of the method on rainfall with different time resolutions, but also reflect the spatiotemporal prediction capabilities of the method.

### 4.3. Experimental Results and Analysis

We try three deep learning model input methods: rainfall data single input, radar echo data single input, and rainfall data and radar echo data dual input. The single input method of rainfall data is similar to the radar echo extrapolation method, which forecasts its future development by past data. The single input method of radar echo data are used to detect whether the radar echo data are effective for rainfall nowcasting. The dual-input model can simultaneously obtain the meteorological spatiotemporal feature of radar echo data and rainfall data. Three deep learning models are tried for each input method, including ConvLSTM, TrajGRU, and PredRNN. The experiment contains nine deep learning methods, and the best method is dual-input dual-encoder RNN which uses TrajGRU, namely RN-Net.

In addition, to compare with the traditional method, the WRF model is run to get the rainfall nowcasting. Its spatial scope is the same as the dataset, and its time scope covers the testing set of the dataset. The WRF model is integrated every 6 h and forecasts the next 12 h with a time resolution of 0.5 h of rainfall. Our deep learning methods forecast the rainfall for the next 2 h every 0.5 h. In order to compare the two types of rainfall nowcasting methods, we designed a special comparison method, as shown in Figure 3. First, we extract 2 h of data every 0.5 h from the 12 h WRF model rainfall forecast, and a total of 21 sets of data. Then, we compare each set of data with the corresponding real rainfall. However, the WRF model has a spin-up period whose duration cannot be determined, and the forecast in this period is usually not used. In order to avoid the spin-up period, the best evaluation results among the 21 sets of data are used as the WRF model evaluation result within 12-h. Meanwhile, we also compared our deep learning methods rainfall forecast of these 21 time periods with the corresponding real rainfall. The average of the 21 evaluation results is used as our deep learning methods evaluation result within 12-h. We compare all the 12-h WRF model forecasts integrated every 6 h in the testing set with the deep learning methods forecasts through the above method. This comparison method not only solves the problem of the different forecasting frequencies of the two methods, but also avoids the spin-up period of the WRF model.

The evaluation results of cumulative rainfall nowcasting within 0.5 h, 1 h, and 2 h are shown in Table 4, Table 5 and Table 6, respectively. The average values of multi-frame 0.5 h rainfall forecast evaluation results within 1 h and 2 h are shown in Table 7 and Table 8, respectively. When comparing the performance of rainfall nowcasting methods, TS and CSI is the main basis. RN-Net has the highest TS and CSI in Table 4, Table 5, Table 6, Table 7 and Table 8 among all rainfall nowcasting methods. Next, we mainly compare the evaluation results of different methods from the following three aspects.

**Comparison between Deep Learning Methods and WRF Model:** As shown in Table 4, Table 5, Table 6, Table 7 and Table 8, the deep learning methods are better than the WRF model. Compared with WRF model rainfall nowcasting, RN-Net’s TS within 0.5 h, 1 h, and 2 h of 0.25 mm as the threshold are increased by nearly four times, three times, and three times, respectively, and RN-Net’s CSI within 1 h and 2 h of 0.25 mm as the threshold are increased by nearly four times and three times, respectively. The rainfall nowcasting in the WRF model is not effective and the FAR is extremely high. This result is caused by two factors. First, it does not use the latest truth data, but is completely dependent on WRF simulations, and WRF simulations usually have deviations in the time domain and geographical area. Second, these methods are manually designed by meteorological experts and can hardly benefit from a large amount of historical data. In addition, the effect of WRF model at low time resolution is better than that at high time resolution, while the deep learning methods are the opposite. This is because the deep learning methods use high time resolution data as training data.

**Comparison of Different Input Methods:** The rainfall extrapolation method that is similar to the radar echo extrapolation method is not good, and even worse than methods that use radar echo data as input. This is due to the fact that spatiotemporal features of the rainfall data are too small to support their forecasts. The dual-input methods are better than the single-input methods.

**Comparison of Different Network Components:** PredRNN, which has the best performance in the three radar echo extrapolation models, has the worst effect in rainfall nowcasting. This is because the scale and quality of the dataset are not enough to support the training of complex deep learning models. The method that uses PredRNN will be more prone to overfitting during the training. Moreover, although the method using TrajGRU is superior to the method using ConvLSTM on TS/CSI and POD, it is slightly inferior to the latter on MSE, MAE, and FAR. This is due to the mechanism of TrajGRU to generate the state connection structure, and this needs further improvement.

### 4.4. Visualization Results

Figure 4 visualizes two representative cases for RN-Net, RE-RF-ConvLSTM, RE-RF-PredRNN, RE-TrajGRU, RF-TrajGRU, and WRF model. From this figure, we observe that all of the deep learning methods except RF-TrajGRU can make accurate forecasting in the first hour, which is consistent with the performance of evaluation metrics. The rainfall data which is input data of RF-TrajGRU are not enough to provide enough meteorological spatiotemporal features to support rainfall nowcasting. Even in the first half-hour, there are some deviations. Moreover, the forecast results of the deep learning methods will gradually disappear after one hour. Among the three dual-input methods, the disappearance problem of the method including ConvLSTM is the most serious. This is due to the structure of RNN [30] and the distance loss function [31]. However, there is no identical situation with the WRF model. WRF model forecast results often have large-scale false alarms, which is identical to the performance in evaluation metrics.

## 5. Conclusions

In this paper, we propose a model (namely RN-Net) for rainfall nowcasting. RN-Net is a deep neural network with dual-input and dual-encoder. RN-Net provides a more sufficient basis for forecasting by fusing the spatiotemporal features of rainfall data and radar data. On the one hand, it overcomes the drawback of conventional forecasting methods that cannot mine knowledge from historical data. On the other hand, it provides high spatiotemporal resolution forecasting that other deep learning methods cannot achieve. We conduct experiments on the Southeastern China dataset. In the experiment, RN-Net is much better than the WRF model. However, compared with the accuracy of precipitation intensity nowcasting [14,15,17,18,20], RN-Net’s rainfall nowcasting still has room for improvement. Moreover, the generalization ability of RN-Net may be poor. This is because rainfall is affected by topography, climate, season, and other factors, and our dataset only contains summer and autumn data of Southeast China.

In order to further improve the accuracy and generalization of rainfall nowcasting, we will extend our current work to three aspects. Firstly, we will increase the scale of the dataset and expand the area included in the dataset. Secondly, we will add more input data to provide more meteorological spatiotemporal features for forecast. Finally, we look forward to future work trying other novel deep learning networks.

## Figures and Tables

**Figure 1 sensors-21-01981-f001:**
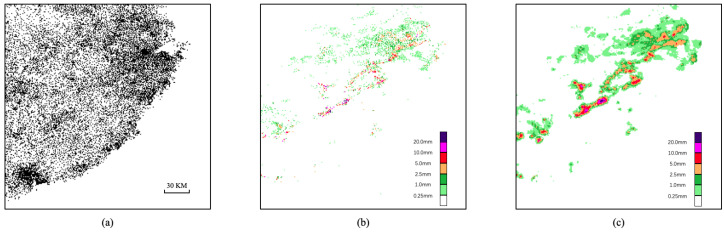
(**a**) Distribution map of automatic weather stations; automatic weather station rainfall data of 9:00 a.m.–9:30 a.m. on 23 May 2017; (**b**) and data after interpolation (**c**).

**Figure 2 sensors-21-01981-f002:**
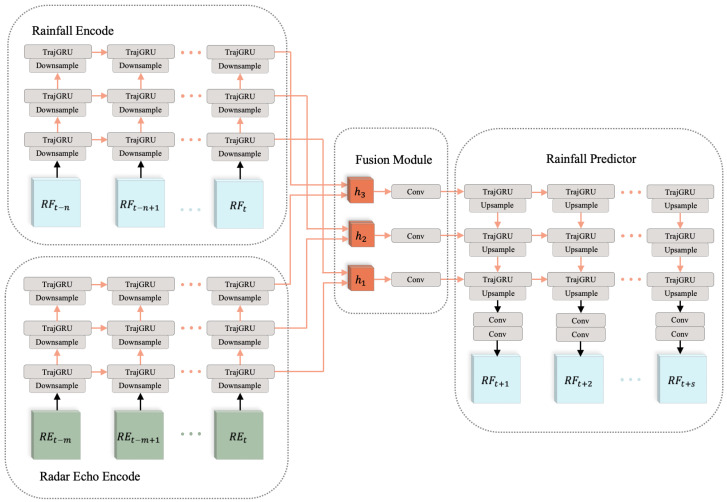
RN-Net consists of four parts: Radar Echo Encoder, Rainfall Encoder, Fusion module, and Rainfall Predictor. The Radar Echo Encoder and the Rainfall Encoder encode spatiotemporal features of radar echo data ${\left[RE_{t}\right]}_{t=-n}^{0}$ and recent rainfall data ${\left[RF_{t}\right]}_{t=-m}^{0}$, respectively. Then, the fusion module combines the radar echo feature and rainfall feature so as to provide more spatiotemporal feature support for nowcasting. Finally, the rainfall predictor receives the fused feature and makes forecasts ${\left[{\hat{RF}}_{t}\right]}_{t=1}^{s}$.

**Figure 3 sensors-21-01981-f003:**
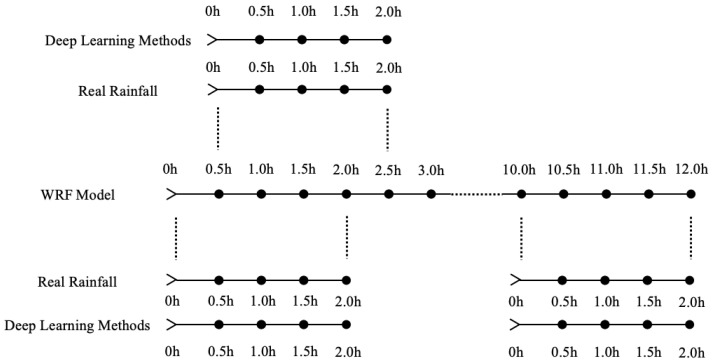
Schematic diagram of comparison method between our deep learning methods and the WRF model.

**Figure 4 sensors-21-01981-f004:**
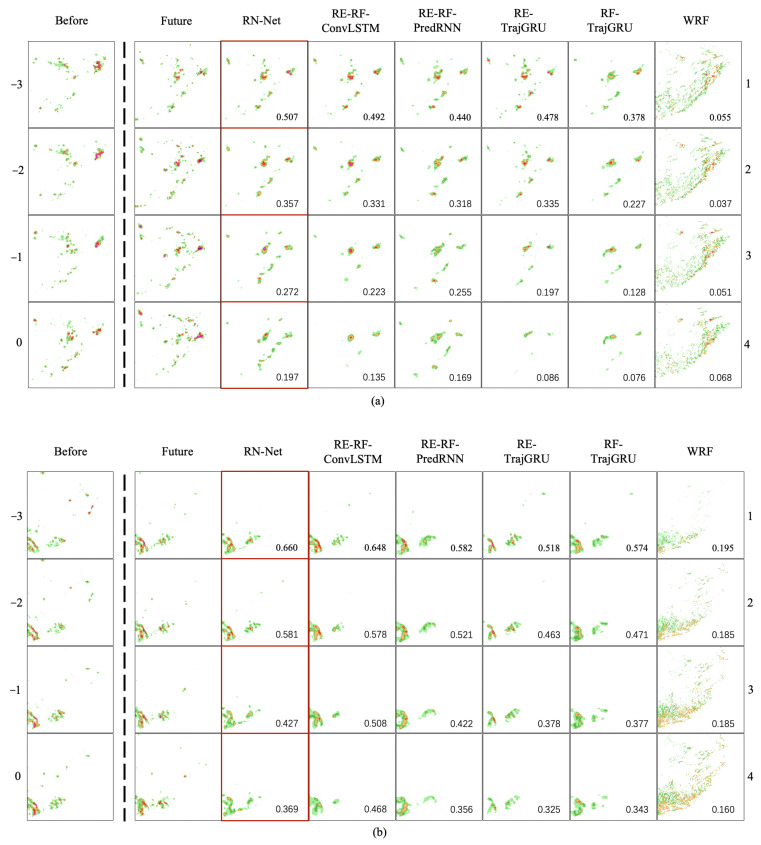
Visualize two representative rainfall nowcasting cases. In (**a**,**b**), from left to right, are the actual rainfall data of the past two hours ${\left[{RF}_{t}\right]}_{t=-3}^{0}$, the actual rainfall data of the next two hours ${\left[{RF}_{t}\right]}_{t=1}^{4}$ and rainfall nowcasting ${\left[{\hat{RF}}_{t}\right]}_{t=1}^{4}$ made by RN-Net, RE-RF-ConvLSTM, RE-RF-PredRNN, RE-TrajGR, RF-TrajGRU, and WRF models. The value in each forecast frame is the CSI with 0.25 mm as the threshold for this frame.

**Table 1 sensors-21-01981-t001:** Physical parameterization schemes.

Name	Scheme
Microphysics	Thompson scheme
Cumulus parameterization	Kain–Fritsch (new Eta) scheme
Planetary boundary layer	Mellor–Yamada–Janjic TKE scheme
Surface layer	Revised MM5 Monin–Obukhov scheme
Longwave radiation	Rapid Radiative Transfer Model for GCMs
Shortwave radiation	Rapid Radiative Transfer Model for GCMs

**Table 2 sensors-21-01981-t002:** Various settings in RN-Net, including channels, kernel, and stride.

Module	Name	CH I/O	Kernel	Stride
Radar Echo Encode or Rainfall Encode	Econv1 ETrajGRU1 Econv2 ETrajGRU2 Econv3 ETrajGRU3	1/8 8/64 64/192 192/192 192/192 192/192	5 × 5 3 × 3 4 × 4 3 × 3 3 × 3 3 × 3	3 1 2 1 3 1
Fusion Module	Fconv1 Fconv2 Fconv3	128/64 384/192 384/192	3 × 3 3 × 3 3 × 3	1 1 1
Rainfall Predictor	PTrajGRU3 Pdeconv3 PTrajGRU2 Pdeconv2 PTrajGRU1 Pdeconv1 Oconv1 Oconv2	192/192 192/192 192/192 192/64 64/64 64/8 8/8 8/1	3 × 3 3 × 3 3 × 3 4 × 4 3 × 3 5 × 5 3 × 3 1 × 1	1 3 1 2 1 3 1 1

**Table 3 sensors-21-01981-t003:** Correspondence between threshold and rainfall level.

Half-Hour of Rainfall (mm)	Rainfall Level
r < 0.25	No or hardly noticeable
0.25 ≤ r < 1	Light
1 ≤ r < 2.5	Light to moderate
2.5 ≤ r	Moderate or greater

**Table 4 sensors-21-01981-t004:** Evaluation results of 0.5 h cumulative rainfall nowcasting. The best performance is reported using red, and the second best is reported using blue. ‘↑’ means that the higher the score the better, while ‘↓’ means that the lower score the better. ‘r≥γ’ means the skill score at the γ mm rainfall threshold in 0.5 h. RE, RF, and RE-RF, respectively, indicate that the method uses radar echo data, rainfall data, or both as input data. PredRNN, ConvLSTM, and PredRNN are RNN units or backbone networks used in this method.

Method	MSE↓	MAE↓	r ≥ 0.25 mm TS↑ POD↑ FAR↓	r ≥ 1 mm TS↑ POD↑ FAR↓	r ≥ 2.5 mm TS↑ POD↑ FAR↓
WRF	1.077	26.602	0.131 0.359 0.763	0.094 0.278 0.830	0.066 0.190 0.868
RF-PredRNN	0.965	19.793	0.404 0.495 0.311	0.303 0.348 0.298	0.216 0.243 0.336
RF-ConvLSTM	0.890	18.889	0.425 0.506 0.272	0.348 0.409 0.299	0.268 0.310 0.337
RF-TrajGRU	0.929	19.024	0.427 0.516 0.287	0.326 0.378 0.293	0.235 0.265 0.324
RE-PredRNN	1.106	22.750	0.380 0.380 0.344	0.344 0.361 0.389	0.189 0.228 0.475
RE-ConvLSTM	0.932	20.550	0.428 0.501 0.254	0.354 0.412 0.285	0.261 0.301 0.334
RE-TrajGRU	0.872	20.514	0.455 0.546 0.268	0.400 0.492 0.318	0.315 0.392 0.383
RE-RF-PredRNN	0.841	18.517	0.474 0.585 0.285	0.393 0.478 0.310	0.296 0.356 0.360
RE-RF-ConvLSTM	0.674	16.531	0.507 0.577 0.193	0.452 0.519 0.221	0.323 0.446 0.266
RN-Net	0.698	16.484	0.523 0.611 0.214	0.464 0.551 0.252	0.371 0.433 0.278

**Table 5 sensors-21-01981-t005:** Evaluation results of 1 h cumulative rainfall nowcasting. ‘r≥γ’ means the skill score at the γ mm rainfall threshold in 1 h.

Method	MSE↓	MAE↓	r ≥ 0.25 mm TS↑ POD↑ FAR↓	r ≥ 1 mm TS↑ POD↑ FAR↓	r ≥ 2.5 mm TS↑ POD↑ FAR↓
WRF	3.635	53.311	0.153 0.419 0.736	0.129 0.363 0.759	0.098 0.294 0.818
RF-PredRNN	3.104	40.523	0.395 0.454 0.245	0.327 0.295 0.280	0.249 0.278 0.261
RF-ConvLSTM	2.963	39.124	0.424 0.491 0.244	0.335 0.374 0.237	0.274 0.304 0.262
RF-TrajGRU	3.066	39.811	0.428 0.503 0.255	0.331 0.380 0.276	0.258 0.291 0.302
RE-PredRNN	3.515	46.225	0.352 0.411 0.287	0.325 0.398 0.358	0.247 0.297 0.407
RE-ConvLSTM	3.083	42.151	0.365 0.395 0.170	0.362 0.407 0.235	0.235 0.320 0.261
RE-TrajGRU	2.818	41.777	0.423 0.508 0.283	0.411 0.486 0.272	0.356 0.430 0.324
RE-RF-PredRNN	2.713	38.258	0.454 0.525 0.227	0.411 0.494 0.290	0.340 0.407 0.325
RE-RF-ConvLSTM	2.260	34.573	0.465 0.508 0.154	0.442 0.489 0.177	0.392 0.439 0.214
RN-Net	2.358	34.959	0.5030.585 0.217	0.461 0.533 0.226	0.3990.467 0.266

**Table 6 sensors-21-01981-t006:** Evaluation results of 2 h cumulative rainfall nowcasting. ‘r≥γ’ means the skill score at the γ mm rainfall threshold in 2 h.

Method	MSE↓	MAE↓	r ≥ 0.25 mm TS↑ POD↑ FAR↓	r ≥ 1 mm TS↑ POD↑ FAR↓	r ≥ 2.5 mm TS↑ POD↑ FAR↓
WRF	12.467	130.027	0.168 0.466 0.742	0.160 0.422 0.758	0.132 0.384 0.780
RF-PredRNN	10.196	87.180	0.341 0.380 0.230	0.295 0.333 0.278	0.235 0.261 0.295
RF-ConvLSTM	10.018	84.924	0.352 0.391 0.217	0.277 0.297 0.194	0.227 0.241 0.202
RF-TrajGRU	10.262	87.172	0.360 0.409 0.247	0.285 0.319 0.275	0.228 0.254 0.310
RE-PredRNN	11.308	97.919	0.315 0.368 0.313	0.297 0.361 0.373	0.244 0.293 0.407
RE-ConvLSTM	10.635	91.343	0.293 0.309 0.148	0.289 0.310 0.193	0.239 0.256 0.219
RE-TrajGRU	9.655	90.043	0.378 0.441 0.274	0.359 0.413 0.266	0.318 0.372 0.313
RE-RF-PredRNN	9.167	83.913	0.399 0.461 0.250	0.3680.440 0.307	0.314 0.373 0.337
RE-RF-ConvLSTM	8.282	77.884	0.393 0.421 0.144	0.368 0.394 0.154	0.325 0.351 0.183
RN-Net	8.465	79.188	0.4350.495 0.218	0.3960.453 0.242	0.3500.406 0.283

**Table 7 sensors-21-01981-t007:** Average evaluation results of two frames of rainfall nowcasting in 1 h. ‘r≥γ’ means the skill score at the γ mm rainfall threshold in 0.5 h.

Method	MSE↓	MAE↓	r ≥ 0.25 mm CSI↑ POD↑ FAR↓	r ≥ 1 mm CSI↑ POD↑ FAR↓	r ≥ 2.5 mm CSI↑ POD↑ FAR↓
WRF	1.223	27.484	0.129 0.355 0.784	0.092 0.285 0.846	0.065 0.190 0.883
RF-PredRNN	1.091	21.201	0.349 0.424 0.339	0.244 0.280 0.349	0.163 0.183 0.405
RF-ConvLSTM	0.890	20.424	0.347 0.404 0.276	0.264 0.303 0.311	0.189 0.214 0.365
RF-TrajGRU	1.076	20.785	0.354 0.425 0.322	0.257 0.297 0.353	0.176 0.198 0.405
RE-PredRNN	1.217	24.064	0.337 0.428 0.390	0.241 0.298 0.447	0.144 0.173 0.540
RE-ConvLSTM	1.086	21.913	0.371 0.431 0.273	0.281 0.322 0.309	0.189 0.214 0.371
RE-TrajGRU	1.013	22.023	0.407 0.490 0.295	0.342 0.423 0.361	0.257 0.318 0.435
RE-RF-PredRNN	0.986	20.351	0.416 0.522 0.331	0.326 0.401 0.372	0.232 0.280 0.433
RE-RF-ConvLSTM	0.840	18.321	0.446 0.508 0.219	0.377 0.433 0.259	0.303 0.350 0.308
RN-Net	0.867	18.599	0.456 0.539 0.254	0.3850.462 0.312	0.289 0.340 0.356

**Table 8 sensors-21-01981-t008:** Average evaluation results of four frames of rainfall nowcasting in 2 h. ‘r≥γ’ means the skill score at the γ mm rainfall threshold in 0.5 h.

Method	MSE↓	MAE↓	r ≥ 0.25 mm CSI↑ POD↑ FAR↓	r ≥ 1 mm CSI↑ POD↑ FAR↓	r ≥ 2.5 mm CSI↑ POD↑ FAR↓
WRF	1.514	35.190	0.124 0.357 0.809	0.084 0.296 0.873	0.062 0.190 0.898
RF-PredRNN	1.270	23.446	0.273 0.331 0.398	0.174 0.199 0.441	0.107 0.119 0.519
RF-ConvLSTM	1.246	22.634	0.249 0.282 0.286	0.175 0.197 0.343	0.113 0.126 0.447
RF-TrajGRU	1.271	23.345	0.263 0.313 0.367	0.180 0.208 0.459	0.115 0.129 0.533
RE-PredRNN	1.384	26.327	0.268 0.347 0.475	0.171 0.211 0.549	0.092 0.109 0.650
RE-ConvLSTM	1.303	24.053	0.269 0.307 0.302	0.180 0.202 0.348	0.109 0.122 0.431
RE-TrajGRU	1.239	24.555	0.320 0.385 0.359	0.250 0.307 0.448	0.171 0.210 0.537
RE-RF-PredRNN	1.198	23.227	0.331 0.423 0.415	0.239 0.295 0.472	0.152 0.182 0.547
RE-RF-ConvLSTM	1.093	21.132	0.342 0.387 0.262	0.270 0.307 0.323	0.201 0.230 0.391
RN-Net	1.118	21.859	0.3550.422 0.333	0.2770.335 0.418	0.1900.223 0.489

## Data Availability

The raw/processed data required to reproduce these findings cannot be shared at this time as the data also forms part of an ongoing study.

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
