# Peer review of "RN-Net: A Deep Learning Approach to 0–2 Hour Rainfall Nowcasting Based on Radar and Automatic Weather Station Data"

_sensors, 2021, doi:10.3390/s21061981_

Round 1

Reviewer 1 Report

Authors proposed RN-Net for a rainfall nowcasting. Reviewer has some concerns about comparison and discussion of the numerical results, and the data used in the manuscripts. Please check the attached documents for more comments.

Author Response

Thank you very much for your question. After revising your question, our article has become more reasonable and more accurate.

The responses to the questions are included in the attachment(Revision1.docx).

Reviewer 2 Report

see attached file

Author Response

The responses to the questions are included in the attachment(Revision2.docx).

Round 2

Reviewer 1 Report

I appreciate authors preparing revision of the manuscript. Authors carefully answered reviewer’s questions and comments. In the previous manuscript, I misunderstood the paper, but now the manuscript became clear enough to understand its novelty. I found some typos to be corrected.

Typos:

  • 2, L. 62: “To solve this problem, We”
  • Eq (5) and (6): authors write “Radar minus Echo minus Encoder” but is it “Radar “hyphen” Echo “hyphen” Encoder” or RE “hyphen” Encoder? In tex, you can use $\mathchar`-$ for hyphens in math mode.

Author Response

Thank you very much for your comments and suggestions .

The responses to the questions are included in the attachment(Revision1.docx).

Reviewer 2 Report

The conclusion could have been substantially improved as had been previously recommended 

Author Response

Thank you very much for your comments and suggestions .

The responses to the question are included in the attachment(Revision2.docx).
